# An Uninvited Seat at the Dinner Table: How Apicomplexan Parasites Scavenge Nutrients from the Host

**DOI:** 10.3390/microorganisms9122592

**Published:** 2021-12-15

**Authors:** Federica Piro, Riccardo Focaia, Zhicheng Dou, Silvia Masci, David Smith, Manlio Di Cristina

**Affiliations:** 1Department of Chemistry, Biology and Biotechnology, University of Perugia, 06122 Perugia, Italy; federica.piro@unipg.it (F.P.); riccardo.focaia@studenti.unipg.it (R.F.); silvia.masci@studenti.unipg.it (S.M.); 2Department of Biological Sciences, Clemson University, Clemson, SC 29634, USA; zdou@clemson.edu; 3Department of Disease Control, Moredun Research Institute, Midlothian EH26 0PZ, UK; d.smith@moredun.ac.uk

**Keywords:** Apicomplexa, *Plasmodium*, *Toxoplasma*, transporters, channels, pumps, carriers, nutrients, parasitophorous vacuole, plasma membrane, digestive vacuole, vacuolar compartment

## Abstract

Obligate intracellular parasites have evolved a remarkable assortment of strategies to scavenge nutrients from the host cells they parasitize. Most apicomplexans form a parasitophorous vacuole (PV) within the invaded cell, a replicative niche within which they survive and multiply. As well as providing a physical barrier against host cell defense mechanisms, the PV membrane (PVM) is also an important site of nutrient uptake that is essential for the parasites to sustain their metabolism. This means nutrients in the extracellular milieu are separated from parasite metabolic machinery by three different membranes, the host plasma membrane, the PVM, and the parasite plasma membrane (PPM). In order to facilitate nutrient transport from the extracellular environment into the parasite itself, transporters on the host cell membrane of invaded cells can be modified by secreted and exported parasite proteins to maximize uptake of key substrates to meet their metabolic demand. To overcome the second barrier, the PVM, apicomplexan parasites secrete proteins contained in the dense granules that remodel the vacuole and make the membrane permissive to important nutrients. This bulk flow of host nutrients is followed by a more selective uptake of substrates at the PPM that is operated by specific transporters of this third barrier. In this review, we recapitulate and compare the strategies developed by Apicomplexa to scavenge nutrients from their hosts, with particular emphasis on transporters at the parasite plasma membrane and vacuolar solute transporters on the parasite intracellular digestive organelle.

## 1. Introduction

The phylum Apicomplexa comprises a large group of obligate intracellular parasites accounting for more than 6000 species. Four classes of Apicomplexa, (*Hemosporidia*, *Coccidia*, *Cryptosporidia,* and *Piroplasmida*) comprise species that are responsible for serious human and animal diseases of medical, veterinary, and economic importance. These include *Plasmodium* spp., *Cryptosporidium* spp., *Toxoplasma gondii*, *Neospora* spp., *Eimeria* spp., *Babesia* spp., and *Theileria* spp. [1]. The life cycles of apicomplexan parasites can be very complex [2]. In fact, while some apicomplexans are monoxenous (life cycle restricted to a single host, e.g., *Cryptosporidium* and *Eimeria*) or dixenous (two successive hosts are required to complete sexual reproduction, e.g., *Plasmodium*, *Babesia*, and *Theileria*), few have the ability to infect multiple hosts (polyxenous). For example, *T. gondii* only completes sexual reproduction in the Felidae family but is capable of asexual replication in any warm-blooded animal worldwide, thus making this parasite one of the most successful pathogens on the planet [3]. The diversity in the host range parasitized by apicomplexans has been driven by the evolution of different strategies that permit survival within diverse environmental niches. A good example of one of these adapted strategies is the way they scavenge nutrients from a host. For example, the asexual erythrocytic stages of *Plasmodium* exploit the large quantity of hemoglobin contained in blood cells as the main source of amino acids for protein synthesis, acquired by endocytosing a large proportion of the host cell cytosol [4]. Differently, *Cryptosporidium* infection is restricted to host gut epithelial cells, where the parasite occupies an intracellular niche at the brush border. Here, a convoluted set of membrane invaginations making up the feeder organelle (unique to *Cryptosporidium* spp.) is generated at the interface between the parasite and the host cell, across which the parasite extracts nutrients from the host cell (e.g., glucose and galactose derived from lactose, the main energy source for suckling mammals) [5]. The cosmopolitan *T. gondii* is instead capable of parasitizing a large variety of nucleated cells of the infected host, reflecting its ability to exploit many different niches to obtain nutrients and sustain itself [6]. This protozoan, and others among the cyst-forming coccidia, are able to survive long-term within the host by taking up residence in long-lived cells, such as neurons and myocytes.

Apicomplexan parasites have lost many anabolic pathways and have instead evolved a remarkable assortment of strategies to scavenge nutrients from their host from within their intracellular residence, establishing huge diversification of host–parasite interactions. These organisms progress through very different ecological niches throughout their life cycle, including different host species and the environment [2]. These distinct niches are profoundly different concerning the availability of specific nutrients; thus, developing strategies to efficiently internalize available and potentially limited substrates (such as glucose and amino acids) from the extracellular milieu is essential to meet the metabolic needs of the parasites and ultimately permit their survival. Most of the nutrients are not permeable to cell membranes and require specific transporters to reach the cell sub-compartment(s) where they enter specific metabolic processes for energy production and protein, lipid, carbohydrate, and nucleic acid synthesis and uptake. Nutrient transporters account for a large number of transmembrane proteins that can be grouped into two classes: “channels” and “carriers” [7]. Channels and passive transporters (also known as facilitated transporters) are characterized by a passive flow of substrates down concentration gradients, while carriers undergo conformational changes after substrate binding to move molecules across membranes against their concentration, with a specific substrate quantity transported per each translocation cycle. Carriers are classified into active and secondary active transporters [8], depending on whether they utilize the energy of ATP binding and hydrolysis to facilitate in- or efflux of their substrates across membranes (active carrier transporters) [9,10,11,12], or whether they exploit ion electrochemical gradients, such as sodium or proton gradients, for the uphill transport of the substrates (secondary active transporters) [13]. Table 1 summarizes transporter types and their most important features [14,15,16,17,18,19,20,21,22,23,24,25,26,27,28,29,30,31,32,33,34,35].

The intense work to discover and characterize the strategies employed by eukaryotic cells to uptake nutrients from the extracellular milieu has also been extended to intracellular parasites, as understanding of these processes may lead to the development of new therapeutic strategies and targets. Some of these organisms reside within the cytosol of the parasitized host cells, with their own plasma membranes as the only impediment to utilize host nutrients. However, apicomplexan parasites, with few exceptions, are instead separated from the host cytosol by a membrane-wrapped compartment, termed the parasitophorous vacuole (PV), where these organisms live and proliferate [36,37]. Although this exclusive environment protects the parasites against infected host cell defenses, it also creates an additional barrier to host nutrients (in addition to the host cell membrane and the parasite cell membrane). In this review, we discuss how intracellular apicomplexan parasites overcome these barriers and have access to host nutrients. Most of our knowledge on the molecular mechanisms employed by these parasites to scavenge host nutrients is confined to *Plasmodium spp.* and *T. gondii*, and thus this review recapitulates and compares these processes in these two important human pathogens.

## 2. Apicomplexa Nutrient Uptake

### 2.1. Scavenging Nutrients across a Physical Barrier—The PVM

Apicomplexans have an intracellular life stage in which they either reside and replicate within a host membrane-derived parasitophorous vacuole that is within the host cell itself (e.g., *Plasmodium* spp., *Cryptosporidium* spp., and *T. gondii* [2]), or replicate freely in the cytoplasm (e.g., *Theileria* and *Babesia* [38]). Residency within the PV represents a parasite defense strategy that confers resistance against innate immune responses of the parasitized host cell. However, the creation of a separate environment from the host cytosol raises a physical barrier that restricts access to host metabolites. To circumvent the problem, parasites such as *Plasmodium* and *T. gondii* remodel their vacuoles to make their membranes permissive to vital substances. Proteins secreted by specialist protozoan organelles (e.g., rhoptries and dense granules) are exported to the PVM and beyond, recruiting host cell mitochondria, the microtubule organizing center (MTOC), and endoplasmic reticulum (ER) to their PVM. This results in the formation of a tubulovesicular network to facilitate the uptake of nutrients from the host [2]. Furthermore, *Plasmodium* heavily modifies the permeability of the erythrocyte plasma membrane by modification of an erythrocyte membrane channel, creating the *Plasmodium* surface anion channel (PSAC), to enhance purine, sugar, anion, amino acid (in particular isoleucine, absent from hemoglobin), and inorganic cation transport across the erythrocyte plasma membrane (Figure 1) [39,40,41,42]. This channel is generated by the new permeability pathways (NPP) induced by internal parasites; however, its molecular composition remains a matter of debate. One model proposes that a CLAG3 dimer/oligomer (part of the family proteins RhopH1) associates with RhopH2 and RhopH3 to form the channel. Alternatively, CLAG3, RhopH2, and RhopH3 may activate an endogenous host channel [43]. Similarly, but using a different mechanism, *Cryptosporidium* enhances the capability of parasitized enterocytes to internalize glucose by upregulating transcription of the host gene GLUT1, although the authors did not address GLUT1 expression at the protein level [44].

Based on “PV-attached” patch-clamp studies in blood-stage *P. falciparum* parasites [45] and passive diffusion of dyes in *T. gondii* [46], it has been demonstrated that the permeability of the PVM allows diffusion of substances less than 1300–1400 Da. This passive diffusion is facilitated by pore-forming protein complexes that do not bind substrates but instead behave like a sieve. *Plasmodium* EXP2 is the pore-forming component of the *Plasmodium* translocon of exported proteins (PTEX) responsible for the export of parasite proteins into the erythrocyte cytosol. However, a PTEX-independent pool of EXP2 is also found at the parasite PVM, forming channels that facilitate the exchange of soluble macromolecules across this membrane. The function of this pore is likely regulated by other parasite proteins, with EXP1 a candidate for this role [47,48,49,50]. Interestingly, pores are similarly present in the PVM of the liver stage parasites [51] and EXP2 expression in this stage [52] indicates that *Plasmodium* blood and liver stages may have a conserved mechanism to diffuse host nutrients across the PVM. A similar channel exists in the *T. gondii* PVM and is composed of two GRA proteins, TgGRA17 and TgGRA23 (Figure 2). These two proteins can be considered orthologs of *Plasmodium* EXP2 owing to a similar predicted tertiary structure and the capacity of EXP2 to rescue TgGRA17 loss of-function phenotypes [46,53]. Using specific dyes, it has been shown that these proteins enable passive, charge-independent, nonselective, and bidirectional diffusion of nutrients and small molecules (up to 1.3 kDa) across the *T. gondii* PVM [46]. Deletion of the TgGRA17 gene in *T. gondii* (Δgra17 strain) results in tachyzoites forming bubble vacuoles with decreased permeability to small molecules. Around 60% of Δgra17 bubble vacuoles eventually rupture, killing the parasites inside. The swollen vacuoles observed in Δgra17 parasites could be explained if TgGRA17 pores were also involved in the elimination of metabolic waste products from the PV lumen, with the absence of these pores resulting in an osmotic imbalance in the PV lumen. Differently, TgGRA23 deletion results in no obvious phenotype, although a double knockout (KO) of TgGRA17 and TgGRA23 was not viable, suggesting that deletion of both genes is a lethal condition for *T. gondii*. TgGRA17-mediated PVM permeability to small molecules is also an important feature for bradyzoite viability, as Δgra17 bradyzoites were significantly less viable compared to wildtype parasites [54]. Each of these *Plasmodium* and *T. gondii* pore-forming proteins lack a transmembrane domain (TM) and have structures that resemble α-pore-forming toxins (αPFTs), suggesting by analogy that they can exist as soluble inactive monomers that can assemble on the PVM to form multimeric pores [55].

The uptake of larger macromolecules across the PVM (>1400 Da) requires other mechanisms. Asexual blood-stage *P. falciparum* parasites utilize the main constituent of erythrocytes, hemoglobin, as a primary source of amino acids for protein synthesis. Erythrocyte hemoglobin is engulfed by an endocytosis apparatus termed the cytostome and the phagotroph, through which *Plasmodium* ingests a large proportion of host cytosol in a process known as the “big gulp” [4]. This is achieved by the coordinated invagination of the PVM and the parasite plasma membrane (PPM) inside the parasite cytosol that eventually fuse with the parasite’s digestive vacuole (DV) [56]. The DV is a lysosomal-like organelle containing several proteases that hydrolyze hemoglobin to release the building blocks for protein synthesis. However, adult human hemoglobin does not contain isoleucine, and thus the parasite must acquire it from the blood plasma via the PSAC and, to a much lesser extent, via an endogenous neutral amino acid carrier (the ‘L system’) [57,58]. Similar to the DV in *Plasmodium*, host cytosolic proteins are also internalized and digested in the lysosomal-like vacuolar compartment (VAC) in *T. gondii* intracellular parasites (recently reviewed in McGovern et al. [59]). Moreover, a study published in 2006 by Coppens et al. [60] proposed a new mechanism by which *T. gondii* parasites gain access to host material, in particular low density lipoproteins (LDL), cholesterol from host endolysosomes. After reorganizing the host cytoskeleton and accumulating host endolysosomes within close proximity to the PVM, invaginations form at the PV membrane that are referred to as host organelle-sequestering tubulo-structures (HOST). These serve as conduits for the transport of host organelles into the vacuolar space. The tubular conduits are coated by the parasite dense granule protein TgGRA7, that works as a ‘‘pinchase’’ constricting the invagination and causing the internalization of HOST within the PV lumen where, after hydrolysis of the membrane organelle, the host sequestered material is absorbed by the parasite [60]. Taken together, the importation of nutrients at the PVM in apicomplexan parasites seems to rely on passive small molecule diffusion via pores or bulk uptake via membrane invaginations. Selective small molecule transport through the PVM is also possible; however, evidence for this is based on only a single identified PVM transporter described in *T. gondii*. This particular transporter is from the ATP-binding cassette subfamily G (ABCG_107_) and is likely involved in lipid uptake [61]. Thus, transport across the PVM is still relatively poorly defined and further studies are required to uncover the full arsenal of PVM transporters exploited by apicomplexan parasites to cross host nutrients into the luminal PV. In particular, most of the limited PVM transport research performed to date has been confined to *Plasmodium* and *Toxoplasma*, and, therefore, nutrient uptake at this site in other apicomplexans remains understudied.

### 2.2. Transport across the Parasite Plasma Membrane

Following uptake into the PV space, another arsenal of transporters is required to internalize nutrients into the individual parasites (Figure 1 and Figure 2 and Table 2). To date, there has been no published comprehensive characterization of this arsenal in the Apicomplexa, and most of the knowledge about these transporters is restricted only to *Plasmodium* and *T. gondii* parasites. In this review, we will only focus on parasite plasma membrane transporters of three large solutes: sugars, nucleotides, and amino acids.

#### 2.2.1. Sugar Transporters

Obligate intracellular pathogens utilize host sugars for energy homeostasis, macromolecular synthesis, and to generate glycoconjugates, all of which are important to their survival and/or virulence. Regarding apicomplexan parasites, glucose metabolism is the major source of carbon and energy for *T. gondii* tachyzoites [83,84,85]. Despite this, glucose is not an essential nutrient for this parasite, as intracellular growth is only reduced by 30% in parasites lacking the major glucose transporter, TgGT1 [62]. TgGT1 KO parasites also showed no differences in in vivo virulence in mice compared to wildtype *T. gondii* parasites. Intracellular growth and survival might still be supported in TgGT1-deficient *T. gondii* through glutaminolysis, releasing pyruvate [62]. Although glycolysis is dispensable for *T. gondii* survival and virulence [86], glucose still represents a major carbon source for many *T. gondii* processes [87,88]. In contrast to *T. gondii*, erythrocytic stages of *Plasmodium* critically depend on glucose uptake, and a continuous supply of glucose is essential for parasite survival [89,90]. Erythrocytic *Plasmodium* increases glucose consumption 50–100 fold as compared to uninfected red cells [91]. Glucose import across the cell membrane of *Plasmodium* or *T. gondii* parasites depends on the hexose transporter PfHT1 and TgGT1, respectively, which are members of the major facilitator superfamily (MFS) [62,63]. While the *Plasmodium* genome has only a single annotated hexose transport gene, *T. gondii* harbors three additional putative sugar transporters (TgST1–3), of which TgST2 is expressed at its surface (whereas TgST1 and TgST3 are intracellular). Ablation of TgST2 causes no phenotype and double KO of TgGT1 and TgST2 is tolerated resulting in a mild growth defect, similar to the single TgGT1 KO [62]. These results indicate that TgGT1 serves as the major hexose transporter at the *T. gondii* plasma membrane. Along with d-glucose, TgGT1 also transports d-mannose, d-galactose, and d-fructose. Likewise, *Plasmodium* PfHT1 exhibits substrate promiscuity, being capable of transporting d-glucose, d-fructose, d-mannose, d-galactose, and d-xylose [92,93]. PfHT1 is also expressed and essential for sugar transport in other stages of *Plasmodium*, as shown by the specific inhibition of the plasmodial hexose transporter in *P. berghei* liver and transmission life stages [94].

Sugar transporters in other Apicomplexa are still poorly characterized. In silico analysis has shown that apicomplexan parasites have at least six distinct phylogenetic subfamilies of sugar transporters [95]. However, excluding *Plasmodium* and *T. gondii*, very little is known on substrate specificity, the timing and location of expression of each member of these subfamilies, and their importance for survival among the different apicomplexan species.

#### 2.2.2. Nucleotide Transporters

*Plasmodium* spp. and *T. gondii* are unable to synthesize purine nucleotides de novo and are therefore reliant upon the salvage of these compounds from the external environment [96,97]. The *P. falciparum* genome contains four putative nucleoside transporters [98], although only one, termed Equilibrative Nucleoside Transporter 1 (PfENT1 or PfNT1), is localized to the parasite plasma membrane and has been characterized functionally [64,65]. PfENT1 is a sodium-independent low-affinity purine transporter with broad specificity for many nucleobases and nucleosides, including adenosine, guanosine, inosine, adenine, guanine, and hypoxanthine [65,67,68,99,100]. PfENT1 is essential for parasite survival, and depletion of this gene is only achievable if purine sources are present at supraphysiological concentrations [101,102,103]. The observation that PfENT1 is essential for parasite survival only at physiological purine levels found in human serum indicates that the parasite plasma membrane must contain alternative transport mechanisms. Frame and collaborators [104] have shown that PfENT4 is also a purine transporter with distinct substrate specificity (such as adenine, adenosine, and 2′-deoxyadenosine) with respect to PfENT1 and, thus, could be an alternative purine transporter that supports the survival of PfENT1-knockout parasites grown in supraphysiological purine conditions. The specific role of PfENT4, as well as PfENT2 and 3, remain to be determined and our knowledge on purine transporters in *Plasmodium* is largely restricted to PfENT1. Interestingly, PfENT2 (termed also PfNT2) is instead an intracellular permease, being localized on the *Plasmodium* endoplasmic reticulum (ER) [66]. This finding raises the possibility that the ER could act as an intracellular purine store. *Plasmodium* auxotrophy for purines represents a potential target for the development of parasite-selective transport inhibitors, particularly because inhibitor specificity and permeant selectivity for purine transporters differ between *Plasmodium* parasites and the host. Encouragingly, inhibitors of the *P. falciparum* ENT1 (PfENT1) have being shown to be very promising for antimalarial drug development [67,105,106].

Similar to *Plasmodium*, *T. gondii* is also auxotrophic for purine [107] and imports these nucleotides in various forms through three transporters, TgAT1, TgAT2, and TgNBT1. TgAT1 (TGME49_244440) is a low-affinity nucleoside transporter for adenosine and inosine [69], while the two uncloned transporters TgAT2 and TgNBT1 are high-affinity purine transporters. TgAT2 is a broad-spectrum nucleoside transporter capable of supporting transport of all natural pyrimidine and purine, while TgNBT1 is a high-affinity purine base transporter that accepts hypoxanthine, xanthine, and guanine as substrates nucleosides [108]. In addition to the TgAT1 gene, three other nucleoside transporter genes are annotated in the ToxoDB database (TGME49_233130, TGME49_288540, and TGGT1_359630). Which one of them encodes the protein responsible for the nucleoside transport activity of TgAT2 or TgNBT1 has not been determined to date.

Again, research into small molecule transport in apicomplexan parasites outside of *Plasmodium* and *T. gondii* remains scarce and, therefore, represents an area that requires further study, particularly given their potential as drug targets.

#### 2.2.3. Amino Acid Transporters

Apicomplexan parasites are auxotrophic for a range of amino acids that must be scavenged from their host cells, thus implying that parasites need to permeate their plasma membrane to gain access to essential amino acids. A family of plasma membrane-localized amino acid transporters have been identified in *T. gondii* genomes and annotated as Apicomplexan Amino acid Transporters (ApiATs) [73]. Members of this family are ubiquitous among the Apicomplexa, and most of them are predicted to be multi-spanning membrane proteins containing 12 TM. The highest sequence similarity was found in the transmembrane domains, and a conserved signature sequence between TM2 and TM3 identified these proteins as members of the major facilitator superfamily (MFS) [73]. The *T. gondii* genome harbors sixteen genes encoding ApiAT proteins that belong to the ApiAT1, ApiAT3, ApiAT5, ApiAT6, and ApiAT7 subfamilies [73]. Eight of these sixteen proteins could be localized in the tachyzoite life stage by tagging into the 3′ end of the open reading frame. Immunofluorescence (IF) microscopy revealed that TgApiAT1, TgApiAT2, TgApiAT3-1, TgApiAT3-2, TgApiAT5-3, TgApiAT6-1 (as reported also previously in [109]), and TgApiAT6-3 all colocalized with the plasma membrane marker SAG1, whilst TgApiAT3-3 showed localization at both the parasite plasma membrane and trans-Golgi network [73]. Although TgApiAT6-2 and TgApiAT7-2 were both detectable by western blotting (WB) of parasite protein extracts, neither of them were detectable by IF microscopy, perhaps due to a low abundance below the limit of detection. TgApiAT5-1, TgApiAT5-2, TgApiAT5-4, TgApiAT5-5, TgApiAT5-6, and TgApiAT7-1 were not detectable by WB and IF microscopy performed on the tachyzoite life stage, and it remains possible that these particular transporters exhibit parasite life stage-specific expression. Single depletion of fifteen ApiAT genes in *T. gondii* showed that only the TgApiAT1, TgApiAT2, and TgApiAT5-3 mutants exhibited defects in tachyzoite growth. The sixteenth gene, TgApiAT6-1, is likely essential for parasite growth since several attempts to inactivate this gene by CRISPR/Cas9 indels resulted in three nucleotides insertion or deletion, without disrupting the reading frame of the gene [73]. This is further supported by a strong negative phenotype score of –5.4 for TgApiAT6-1 from a genome-wide CRISPR/Cas9 screen [110].

TgApiAT1 is reported to be a selective ion-independent arginine transporter essential for parasite survival and virulence [71]. However, TgApiAT1 is dispensable for parasite growth in a medium containing high concentrations of l-arginine, suggesting the presence of at least one other l-arginine transporter. This hypothesis was supported in a recent publication in which TgApiAT6-1 was identified as the transporter that mediates TgApiAT1-independent l-arginine uptake in *T. gondii* [75]. Specifically, two parallel studies on TgApiAT6-1 have demonstrated that this protein is a general cationic amino acid transporter that mediates both the high-affinity uptake of l-lysine and the low-affinity uptake of l-arginine [75,76]. Furthermore, TgApiAT6-1 is the sole l-lysine transporter in *T. gondii*, explaining the indispensability reported in the corresponding gene [76]. TgApiAT6-1 has a broad specificity for many cationic and large neutral amino acids and l-arginine metabolites. However, its very high selectivity for l-lysine reduces the transport rate for all other substrates when l-lysine is present. Both TgApiAT6-1 and TgApiAT1 transporters function as bidirectional uniporters and are capable of substrate exchange. The uptake of cationic amino acids by TgApiAT6-1 is ‘trans-stimulated’ by cationic and neutral amino acids, while TgApiAT1 is solely trans-stimulated by l-arginine. Parasite intracellular uptake of l-lysine and l-arginine against their gradient may be favored by the negative inside membrane potential (Em) that may be present across the plasma membrane of *T. gondii* intracellular parasites. Alternatively, the constant consumption of imported cationic amino acids for parasite protein synthesis and metabolism may maintain a low intracellular concentration, favoring the uptake. The abundance of TgApiAT1 is regulated by the amount of available l-arginine in the parasite’s external environment, increasing in response to decreased l-arginine [74]. Fittingly, the TgApiAT1 protein level has been shown to vary in parasites infecting different host organs depending on tissue-specific l-arginine availability. Moreover, TgApiAT1 is also regulated by the availability of other nutrients, such as glucose and the amino acids l-glutamine and l-tyrosine, in an inverse direction compared to the l-arginine response. TgApiAT1 abundance is regulated post-transcriptionally and is mediated by an upstream open reading frame (uORF) in the 5′ untranslated region of the TgApiAT1 transcript. The peptide encoded by this uORF is critical for mediating TgApiAT1 regulation [74].

Taken together, TgApiAT1 abundance is upregulated in response to limited availability of l-arginine but downregulated upon limitation of a range of nutrients such as glucose, l-glutamine, and l-tyrosine, indicating that TgApiAT1 abundance regulation is perhaps part of a more general starvation response. Integrating their findings with research published by Augusto et al. [111], Fairweather et al. [75] proposed an appealing model to explain how an apicomplexan parasite responds to changes in the availability of a key nutrient. Augusto et al. [111] demonstrated that intracellular *T. gondii* induces nutrient starvation in host cells, activating an integrated stress response (ISR) pathway in the infected cells. This is mediated by the phosphorylation of the essential parasite translation factor eIF2 by the protein kinase GCN2. Activated ISR resulted in an increase in the abundance of the mammalian cationic amino acid transporter CAT1 and a subsequent increase in l-arginine and l-lysine uptake into host cells [111]. Thus, scavenging of cationic amino acids by *T. gondii* parasites would result in a diminution of these amino acids in the host cell cytosol causing upregulation of the host CAT1 transporter. Having a similar affinity for l-arginine and l-lysine, this could increase uptake of both amino acids into host cells and thereby maintain the same ratio of these two amino acids either side of the membrane. Depending on which organ *T. gondii* parasites are infecting, the intracellular ratio of l-arginine and l-lysine in infected host cells should vary, being lower in high l-arginine catabolism organs such as the liver and higher in other tissues, for example, within the kidneys that function in the net synthesis of l-arginine. In organs where l-arginine concentration is low compared to l-lysine, l-arginine uptake is outcompeted by l-lysine uptake at the TgApiAT6-1 transporter, resulting in upregulation of TgApiAT1, which enables increased l-arginine uptake for parasite proliferation. If the ratio of l-arginine:l-lysine in the host cell is instead high, l-arginine can compete with l-lysine for uptake by TgApiAT6-1, resulting in a decreased role for TgApiAT1 that translates into downregulation of its abundance. Overall, TgApiAT1 may contribute substantially to l-arginine uptake when l-arginine levels are limited, while TgApiAT6-1 may ensure uptake of l-arginine and l-lysine under nutrient-rich conditions [75].

Parker et al. [73], in parallel to a study published by Wallbank et al. [112], characterized the substrate specificity of another amino acid transporter, TgApiAT5-3. This transporter has a high affinity for l-tyrosine uniporter that does not co-transport any ions and whose depletion leads to defects in l-tyrosine uptake into parasites. Experiments performed using *Xenopus leavis* oocytes have indicated that TgApiAT5-3 can function as an exchanger, with the rate of uptake of l-tyrosine and other aromatic and large neutral amino acids enhanced when equivalent amino acids were present on the trans side of the membrane. Based on this mechanism, TgApiAT5-3 seems to play an important role in maintaining amino acid homeostasis by facilitating both the net uptake of l-tyrosine into the parasite and balancing the intracellular concentrations of aromatic and large neutral amino acids through exchange. Other evidence obtained using *X. leavis* oocytes indicated that TgApiAT5-3 might also function in the net uptake of other amino acids in the parasite such as l-phenylalanine, l-tryptophan, and large neutral amino acids such as l-leucine. However, TgApiAT5-3 affinity for l-phenylalanine is very low compared to that for l-tyrosine; therefore, another transporter might be responsible for the uptake of this amino acid [73]. The authors speculated that another putative l-phenylalanine transporter may be responsible for parasite growth at l-tyrosine concentration of 1 mM providing an alternative l-tyrosine uptake pathway that imports sufficient l-tyrosine to enable parasite growth when a high concentration of l-tyrosine is present. Another interesting finding was that high concentrations of l-phenylalanine or l-tryptophan inhibited the growth of ΔTgApiAT5-3 tachyzoites, even when the cell culture media contained 1 mM l-tyrosine [73]. Parker et al., therefore, proposed a model for l-tyrosine uptake in which the uptake of this amino acid is mediated primarily by TgApiAT5-3, whilst l-phenylalanine and l-tryptophan uptake is primarily mediated by one or more alternative aromatic amino acid transporters that transport l-tyrosine when l-tyrosine levels are high and corresponding levels of l-phenylalanine and l-tryptophan are lower.

Although these recent studies in *T. gondii* have increased our knowledge on amino acid transporters in Apicomplexa, these transporters remain poorly defined and have not been assigned a substrate in most parasites belonging to this phylum. The amino acid demands of *Plasmodium* blood-stage parasites is largely satisfied via the degradation of hemoglobin within the DV. An exception to this is isoleucine, an amino acid *Plasmodium* is auxotrophic for [113,114] and which is not present in adult hemoglobin [115]. Therefore, to gain access to this particular amino acid, it must first be imported from the blood plasma across the host cell membrane into the host cytosol. Methionine is also poorly represented in hemoglobin and some strains of *P. falciparum* are dependent on an exogenous supply of this amino acid [57]. Despite years of investigation to determine the mechanisms for isoleucine and methionine acquisition by *Plasmodium*, they remain elusive. Clues towards the identification of their transporters have come from studies that show that the uptake of these amino acids through the parasite plasma membrane occurs via a carrier that mediates the low-affinity high-capacity transport of neutral amino acids and is independent of H^+^ or Na^+^ [57,58]. The *Plasmodium* transportome has at least eleven genes classified as putative amino acid transporters, and five of them belong to the ApiAT family [98,116,117,118]. The homologue of TgApiAT1 was first identified and characterized in *P. berghei*, initially termed PbNTP1 and then renamed PbApiAT8 [71,72]. This first study showed that PbApiAT8 functions as a cationic amino acid transporter with a broader substrate specificity compared to TgApiAT1. Ablation of PbApiAT8 resulted in significantly impaired microgamete formation and parasite motility in the sexual cycle [72]. In a very recent study [70], localization and essentiality of four ApiAT family members of *P. falciparum* during intraerythrocytic development has been investigated, including the homologue of PbApiAT8, PfApiAT8. Surprisingly, deletion of PfApiAT8 had no impact on gametocyte development and morphology. This discrepancy may be due to the pronounced differences in gametocyte development between *P. falciparum* and *P. berghei* parasites [119]. The other three genes investigated in the same study were PfApiAT2, PfApiAT4, and PfApiAT10. All four PfApiATs were localized on the plasma membrane of asexual blood stage parasites and gametocytes [70]. Individual inactivation of each of these genes found only a minor reduction in parasite growth for PfApiAT2 and PfApiAT4, while the absence of PfApiAT8 and PfApiAT10 caused no apparent phenotype. Although the same study attempted to functionally characterize another transporter, PfApiAT9, the inability to detect protein expression by IF microscopy precluded localization; thus, the gene was not further investigated. An important interpretation of this data is that the lack of any observable major phenotype following the single deletion of these four genes indicates that the transporters they encode for are not individually involved in isoleucine and methionine uptake. However, functional redundancy cannot be excluded from interpretations of the data, either within the ApiAT family or by the presence of yet unassigned transporters capable of transporting isoleucine and methionine across the PPM. The *Plasmodium* transportome includes only three essential transporters in the asexual blood stage known to date, including AA1 (a putative aromatic amino acid transporter), AAAP3 (a member of the amino acid/ auxin permease family), and NSS1 (a putative neurotransmitter:Na^+^ symporter) [77]. The essentiality of these genes would place them in a good position as candidates for being the plasma membrane isoleucine transporter; however, their localization suggests this is not the case. AAT1 and AAAP3 localize to the digestive vacuole membrane and apicoplast, respectively, and the expression profile of NSS1 is not compatible with this function, since its transcription occurs primarily between 27 and 48 h post-invasion [98,118,120,121,122]. In conclusion, the isoleucine transporter remains elusive, and further investigation is required to identify this *Plasmodium’s* Achilles’ heel and potential therapeutic target.

Amino acids liberated via the digestion of relatively large quantities of hemoglobin in the parasite DV need to be released into the parasite cytosol by amino acid effluxers. To date only one putative effluxer has been extensively characterized, namely the Chloroquine Resistance Transporter (CRT). The name of the CRT protein is derived from its function in a mutant form of *P. falciparum*, whereby it pumps chloroquine out of the DV, conferring resistance to this drug [78,79]. Chloroquine has been widely used to control malaria infection as it can penetrate into the acidic DV based on its weak base property, thereby neutralizing the normally acidic pH within the organelle and importantly compromising hydrolysis activity by pH-dependent proteases within this compartment [78]. The orthologs of CRT are widely distributed in many apicomplexan parasites, including *Plasmodium* spp., *T. gondii*, *Eimeria* spp., and *Neospora* spp., although the native function of the CRT orthologues remains unclear. A number of studies have reconstituted recombinant PfCRT isoforms into liposomes or expressed them in frog oocytes to investigate their native substrates [123,124]. The findings of these studies suggest that a series of 4 to 11-residue peptides derived from hemoglobin or other erythrocyte proteins can be transported by PfCRT [124], which coincides with the observation that some hemoglobin-derived peptides accumulate in chloroquine-resistant parasites compared to their wildtype counterparts [125]. In addition, a few small nutrient compounds, such as l-arginine, l-lysine, l-histidine, and glutathione have been reported to be potential substrates of PfCRT via a liposome-based assay [123,126]. Previous research has also indicated that PfCRT might transport iron ions across the digestive vacuole membrane [125]. It is noteworthy that the mutant PfCRT has a fitness defect in that it transports putative native substrates at lower efficiency than the wildtype form [123,124], providing an explanation for the recovery of chloroquine sensitive malaria parasites following the cessation of chloroquine treatment [125].

Compared to PfCRT, the *Toxoplasma* CRT ortholog (TgCRT) can be genetically deleted and the modified strain remains viable [80,81]. Similar to PfCRT, TgCRT is also located in the digestive organelle, in the case of *T. gondii* this is the VAC [80,81,82]. Recent collaborative research between our group and the Carruthers’ laboratory has revealed that *Toxoplasma* ingests and digests host proteins via endocytosis and recycles parasite organelles in the VAC [127,128]. We have shown that although cathepsin L (TgCPL) is dispensable in acute stage tachyzoites [127,128,129], inhibition or genetic ablation of this protease in chronic stage bradyzoites results in parasite death [128,129]. CPL-deficient bradyzoites were also shown to accumulate autophagosomes, indicating autophagy could be essential to chronic persistence [128]. This was corroborated in a following study from Smith et al. [130] that demonstrated that autophagy is constitutively induced in bradyzoites and essential for cyst survival. Inhibition of this process through TgAtg9 gene deletion resulted in bradyzoite death and a dramatic decrease in the brain cyst burden in challenged mice, mirroring the results obtained for the TgCPL knockout strain [128,130]. It still remains to be determined whether autophagy and VAC digestion following encystation are required to renew organelles during the chronic stage (and thus is critical for cellular homeostasis), or if it reflects a starvation response to provide essential substrates to sustain the bradyzoite. In the latter scenario, digested products would need to be exported from the VAC to fuel bradyzoite metabolism. To date, the only potential amino acid effluxer localized to the VAC membrane of *T. gondii* is TgCRT [82]. Although the genetic ablation of TgCRT is achievable, the resulting TgCRT-deficient parasites displayed swollen VACs, approximately 10-fold larger in tachyzoites and more severely bloated in bradyzoites than the corresponding stages in wildtype *Toxoplasma* [80,81]. This is similar to the phenotype observed in chloroquine-resistant *Plasmodium* [78]. Further supporting the role of CRT as a vacuolar effluxer is the finding that this VAC swelling was partially attenuated in the TgCRT knockout strain of *T. gondii* when TgCPL was also deficient; thus, the VAC proteolysis was compromised [80,81]. Furthermore, in the TgCRT-deletion mutant, the swelled VAC was unable to become separated from its adjacent precursor organelle, the endosome-like compartment (ELC) [80], which further altered the physiology within the VAC and compromised its hydrolytic activity. The deletion of TgCRT led to reduced virulence in a murine model for both acute and chronic toxoplasmosis, based on host survival and a lower brain cyst burden [80,81]. We have now identified novel putative solute transporters that are resident to the VAC membrane and are currently underway with their characterization to elucidate their function and solute specificity (unpublished data).

Overall, that bradyzoites replicate and form new daughter parasites creates a paradox for their dependency on autophagy, as autophagy is a recycling process that cannot result in a net increase in material. With this in mind, if autophagy in *T. gondii* bradyzoites does indeed serve a role to supply nutrients, it is reasonable to hypothesize that this is supplemented with material derived from the host cell, implicating a role for cyst wall transporters in nutrient uptake. Whether these cyst wall transporters are the same as those found on the PVM in tachyzoites requires further study. Nevertheless, the essential role for autophagy in bradyzoites would suggest nutrient transport across the cyst wall is not sufficient to sustain parasite survival in this life stage but rather performs a supplementary function.

## 3. Conclusions

The long journey of nutrients from the extracellular milieu to the cytosol of apicomplexan parasites requires passage across multiple membranes that has driven the evolution of different uptake strategies for gaining access to specific solutes. Our knowledge is mainly confined only to two genera in the Apicomplexa, *Plasmodium* and *Toxoplasma*, and even in those parasites it is relatively fragmented (see Table 2). Gene and protein annotation made publically available via the Eukaryotic Pathogen, Vector and Host Informatics Resource (VEuPathDB) is proving to be a powerful tool to identify solute transporters, facilitated by knowledge of transporters from other eukaryotes. That being said, the low amino acid conservation exhibited between apicomplexan transporters and those from other eukaryotes continues to make bioinformatics-based identification of new transporters difficult, particularly from a functional and mechanistic standpoint. Characterization of these proteins is not trivial since most are multi-spanning membrane proteins that can be difficult to purify and to express as recombinants for in vitro studies and antibody production. Uncovering the solute specificity of apicomplexan transporters is still a very difficult and time-consuming process and often requires the use of exogenous systems such as *X. laevis* oocytes, technology that is not readily available to all laboratories. The continued development and application of CRISPR/Cas9 gene editing technology in apicomplexan parasites [131] has been a major advance for studying cell biology in these organisms, including in the interrogation of putative nutrient transporters. This has been particularly true for research performed in *Plasmodium* and *Toxoplasma*, but is now excitingly being well adapted in other apicomplexan parasites too, for example *Cryptosporidium*. The potential for apicomplexan-specific nutrient transporters as drug targets makes these advancements all the more important, as researchers continue to strive to develop new therapeutic interventions against diseases caused by these pathogens of medical and veterinary importance.

## Figures and Tables

**Figure 1 microorganisms-09-02592-f001:**
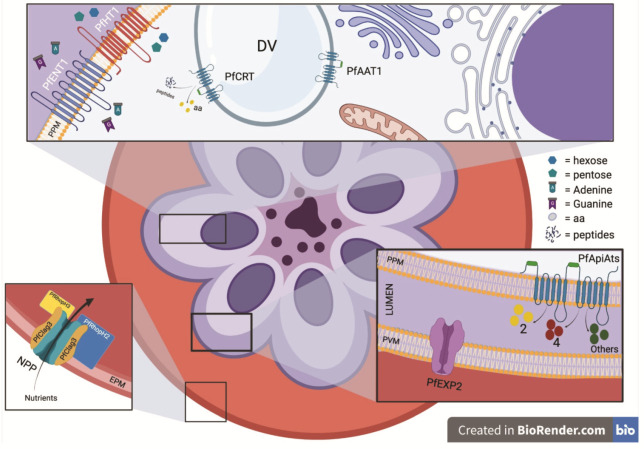
Schematic representation of transporters in a *P. falciparum*-infected erythrocyte. Nutrients in the extracellular milieu must be trafficked across multiple membranes to reach the parasite, including the host cell membrane, the PVM, and the PPM. Known transporters include the following: At the host cell membrane, the PfCLAG3 channel is induced by intracellular parasites through the NPP, facilitating uptake of extracellular nutrients (bottom left inset image). PfENT1 and PfHT1 at the PPM are nucleoside and sugar transporters, respectively (upper inset image). PfApiATs function as amino acid transporters at the PPM (bottom right inset image). PfEXP2 is a pore-forming component of the *Plasmodium* translocon of exported proteins (PTEX) and exports parasite proteins into the erythrocyte cytosol (bottom right inset image). PfAAT1 is an amino acid transporter localized to the DV, while PfCRT exports small molecules from the DV lumen to the parasite cytosol (upper inset image). Aa—amino acid; DV—digestive vacuole; EPM—erythrocyte plasma membrane; NPP—new permeability pathways; PPM—parasite plasma membrane; and PVM—parasitophorous vacuole membrane.

**Figure 2 microorganisms-09-02592-f002:**
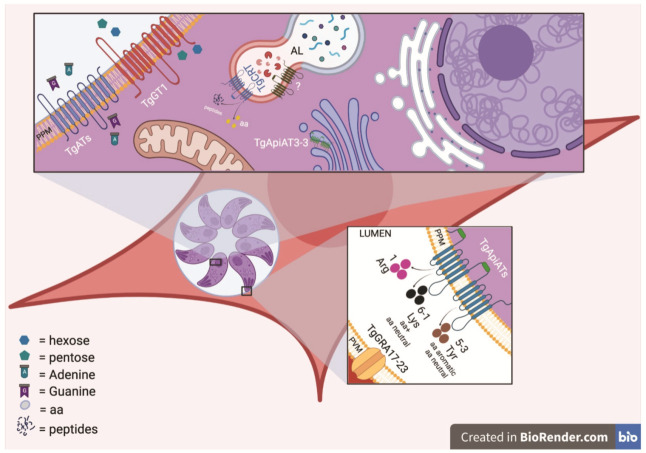
Schematic representation of transporters in intracellular *T. gondii* tachyzoites. As with *P. falciparum*, nutrients must also be trafficked across multiple membranes to reach intracellular *T. gondii* parasites. Known transporters include the following: TgGRA17 and TgGRA23 generate a pore-forming complex on the PVM that facilitates the movement of small molecules across this membrane (lower inset image). Numerous TgApiATs facilitate amino acid transport across the PPM, with different transporters adapted for the movement of specific types of amino acids (lower inset image). TgApiAT3-3 has also been shown to localize to the trans-Golgi network (upper inset image). TgATs and TgGT1 are nucleoside and sugar transporters, respectively, located at the PPM. TgCRT exports small molecules from the plant-like digestive vacuole (the VAC) (upper inset image). No amino acid importer on the VAC has previously been reported. aa—amino acid; aa+—basic amino acid; AL—autophagolysosome; PPM—parasite plasma membrane; and PVM—parasitophorous vacuole membrane.

**Table 1 microorganisms-09-02592-t001:** Schematic overview of Transporters.

Class of Transporters	Sub-Class of Transporters	Porter Type	Energy-Dependent Transport	Substance Transport Direction	Typical Molecules Using Pathway	Refs.
Channels	- α-Type channels - β-Barrel porins - Pore-forming toxins - Non-ribosomally synthesized channels	Potential-dependent channel proteins—activated by a change in the membrane potential Ligand-dependent channel proteins—activated by binding to a ligand-mediator Mechanically dependent channel proteins—activated by mechanical deformation of the cell membrane	NO	Down concentration gradients	Ions: Na^+^, K^+^, Ca^2+^	[14,15]
Active transporters	- ABCs - Pumps: F-type, P-type and V-type ATPases	Importers, exporters and extruders	ATP hydrolysis	Against concentration gradients	Ions: Na^+^, K^+^, Ca^2+^, H^+^	[12,16,17,18,19,20,21,22,23]
Secondary active transporters	SLCs: - MFS - LeuT fold family - NhaA fold family	Symporter (cotransporter) and antiporter (exchanger)	Ion electrochemical gradient (cotransport of Na^+^, H^+^ or Cl^−^ and/or the counter-transport of K^+^)	Against concentration gradients	Polar: amino acids, glucose, some ions	[24,25,26,27,28,29,30,31,32,33,34]
Facilitated transporters	SLCs: - Na^+^-independent Glucose transporters - Na^+^-independent amino acid transporters	Uniporter	NO	Down concentration gradients	Polar: glucose	[25,26,27,35]

Abbreviations—ABC: ATP-binding cassette transporters; SLC: Solute carrier family; MFS: Major facilitator superfamily.

**Table 2 microorganisms-09-02592-t002:** Transporters in *Plasmodium falciparum* and *Toxoplasma gondii*.

	*Toxoplasma gondii*	*Plasmodium falciparum*	ID	Class	Putative Funtion	TM	Nutriens	Size aa	Refs.
Host Cells		PfCLAG3.1	PF3D7_0302500	PSAC	Dimer/oligomer channel form	NO	Purine, sugar, amino acid, inorganic cations.	1417	[43]
		PfCLAG3.2	PF3D7_0302200	PSAC	Dimer/oligomer channel form	NO	Purine, sugar, amino acid, inorganic cations.	1416	[43]
		PfRhopH2	PF3D7_0929400	PSAC	Dimer/oligomer channel form	NO	Purine, sugar, amino acid, inorganic cations.	1378	[43]
		PfRhopH3	PF3D7_0905400	PSAC	Channel form	NO	Purine, sugar, amino acid, inorganic cations.	897	[43]
PVM		PfEXP1	PF3D7_1121600	PTEX	Pore exporting proteins and macromolecules	1	Host nutrients	162	[49,50]
		PfEXP2	PF3D7_1471100	PTEX	Pore exporting proteins and macromolecules	NO	Host nutrients	287	[49,50]
	TgGRA17		TGME49_222170	PTEX Orthologus	Nutrient Transporters	NO	Host nutrients	300	[46,53,54]
	TgGRA23		TGME49_297880	PTEX Orthologus	Nutrient Transporters	1	Host nutrients	219	[46,53]
	TgABCG_107_		TGME49_247540	ATP-binding cassette subfamily G	Lipid import	5	Lipids	981	[61]
PPM		PfHT1	PF3D7_0204700	MSF	Sugar Transporters	12	Sugar	504	[62,63]
	TgGT1		TGME49_214320	MSF	Sugar Transporters	12	Sugar	568	[62,63]
	TgST1		TGME49_257120	MSF	Sugar Transporters	12	Sugar	601	[62]
	TgST2		TGME49_272500	MSF	Sugar Transporters	12	Sugar	689 aa	[62]
	TgST3		TGME49_201260	MSF	Sugar Transporters	10	Sugar	693	[62]
		PfNT1	PF3D7_1347200	PfENT	Nucleotide Transporters	9	Purines	422	[64,65]
		PfNT2	PF3D7_0824400	PfENT	Nucleotide Transporters	10	Purines	585	[66]
		PfNT3	PF3D7_1469400	PfENT	Nucleotide Transporters	11	Purines	437	[65,67,68]
		PfNT4	PF3D7_0103200	PfENT	Nucleotide Transporters	11	Purines	434	[65,67,68]
	TgAT1		TGME49_244440	TgAT Family protein	Nucleotide Transporters	10	Purines	462	[69]
		PfApiAT2	PF3D7_0914700	ApiAT	Amino Acid Transporters	12	?	516	[70]
		PfApiAT4	PF3D7_1129900	ApiAT	Amino Acid Transporters	12	?	609	[70]
		PfApiAT8	PF3D7_0104800	ApiAT	Amino Acid Transporters	12	Catonic Amino Acid	577	[70,71,72]
		PfApiAT10	PF3D7_0312500	ApiAT	Amino Acid Transporters	12	?	579	[70]
	TgApiAT1		TGME49_215490	ApiAT	Amino Acid Transporters	12	Arginine	534	[73,74]
	TgApiAT5-3		TGME49_257530	ApiAT	Amino Acid Transporters	12	Tyrosine, Phenilalanine, Triptofane, Leucine	504	[73]
	TgApiAT6-1		TGME49_240810	ApiAT	Amino Acid Transporters	10	Lysine, Arginine	566	[75,76]
DV/VAC		PfAAT1	PF3D7_0629500	ApiAT	Amino Acid Transporters	9	Aromatic amino acid	606	[77]
		PfCRT	PF3D7_0709000	DMT	Amino Acid Transporters	10	Arginine, Lysine, Histidine, Glutathione	424	[78,79]
	TgCRT		TGME49_313930	DMT	Amino Acid Transporters	9	Amino Acids	881	[80,81,82]

Abbreviations—PVM: Parasitophorous vacuole membrane; PPM: Parasite plasma membrane; DV/VAC: Digestive vacuole/Vacuolar compartment.

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
