# Peer review of "An Uninvited Seat at the Dinner Table: How Apicomplexan Parasites Scavenge Nutrients from the Host"

_microorganisms, 2021, doi:10.3390/microorganisms9122592_

Round 1

Reviewer 1 Report

The manuscript submitted by Piro et al. entitled “An uninvited seat at the dinner table: how apicomplexan parasites scavenge nutrients from the host” is a nicely written piece about the known transport systems responsible for the movement of carbohydrates, nucleotides, and amino acids that have been described to date mainly in two apicomplexan parasites, namely Plasmodium and Toxoplasma. In addition, the transport molecules located in the parasitophorous vacuole were included. They not only allow the capture of nutrients from host cells but, also permit the extrusion of toxic compounds product of parasite metabolism. The significance of those proteins as possible drug targets is highlighted. The authors were cautious and included the most relevant and up-to-date literature. 

The major reflection lies in the introduction. General knowledge of channels and carriers is almost in every biochemistry book accessible. Therefore, I strongly recommend summarizing this information (lines 79 to 124) in a table.   

Minor points:
-    Please delete (PV) for parasitophorous vacuole in line 148. It was defined in line 136.
-    Please correct the spelling “hemoglobulin” in line 163.
-    Line 325, please correct the name TgBT1 to TgNBT1.
-    Please clarify the following sentence: “The genes encoding TgAT2 and TgNBT1 have not been identified to date.” (line 331). This sentence may be misleading. In a quick search on the website ToxoDB, sequences for nucleoside transporters and their localization in the genome were easily found. TGVEG_244440; TGVEG_233130; TGVEG_359630, and TGVEG_288540
-    Reference 129 in the reference list, please delete. It is an iteration of reference 128.

Author Response

We thank the reviewer for taking the time to review our manuscript and for her/his constructive comments.

Point 1: The major reflection lies in the introduction. General knowledge of channels and carriers is almost in every biochemistry book accessible. Therefore, I strongly recommend summarizing this information (lines 79 to 124) in a table.  

Response 1: We agree with the reviewer that there are already many reviews and books comprehensively covering channel and carrier structure and function and then our description can be redundant with the large body of literature on these topics. Following reviewer suggestion, we have dramatically shortened this part of the introduction and generated a new table (Table 1) that summarizes the most important classes of transporters and their features. The table contains references that allow readers to easily find dedicated literature, should they wish to go into details of general classification, structure and function of transporters.

Minor points:

Point 2: Please delete (PV) for parasitophorous vacuole in line 148. It was defined in line 136.

Response 2: We have deleted (PV).

Point 3:    Please correct the spelling “hemoglobulin” in line 163.

Response 3: We have corrected the spelling of hemoglobin.

Point 4:  Line 325, please correct the name TgBT1 to TgNBT1. 

Response 4: We have corrected the name of this transporter.

Point 5:     Please clarify the following sentence: “The genes encoding TgAT2 and TgNBT1 have not been identified to date.” (line 331). This sentence may be misleading. In a quick search on the website ToxoDB, sequences for nucleoside transporters and their localization in the genome were easily found. TGVEG_244440; TGVEG_233130; TGVEG_359630, and TGVEG_288540

Response 5: We thanks the reviewer for raising this point and agree with the reviewer that the way we addressed it was not very clear.

De Koning et al. (1) investigated on the purine uptake systems of T. gondii and identified two

previously unknown high affinity nucleoside transporters. The identification of the presence of these two new transporters was based on biochemical analysis of purine uptake of T. gondii and not at genetic level . The genes encoding these transporters have not been investigated to date. Thus, although three genes have been annotated as nucleoside transporters (along with the fourth gene TgAT1, TGME49_244440), as correctly indicated by the reviewer, there are no experimental evidence of which of them may be responsible for the purine transport activity of TgAT2 and TgNBT1.

To better clarify this point and indicate to the readers that 3 more nucleoside transporters are annotated in ToxoDB database, we have replaced the sentence:

“The genes encoding TgAT2 and TgNBT1 have not been identified to date.” (in the previous line 331)

with: “In addition to the TgAT1 gene, other three nucleoside transporter genes are annotated in the ToxoDB database (TGME49_233130, TGME49_288540 and TGGT1_359630). Which one of them encodes the protein responsible for the nucleoside transport activity of TgAT2 or TgNBT1 has not been determined to date.”

We have also added the ID of the TgAT1 gene, TGME49_244440.

  • De Koning, H.P.; Al-Salabi, M.I.; Cohen, A.M.; Coombs, G.H.; Wastling, J.M. Identification and Characterisation of High Affinity Nucleoside and Nucleobase Transporters in Toxoplasma Gondii. Int J Parasitol 2003, 33, 821–831, doi:10.1016/s0020-7519(03)00091-2.

Point 6:    Reference 129 in the reference list, please delete. It is an iteration of reference 128. 

Response 6: We have deleted this iteration.

Reviewer 2 Report

This is a nice up-to-date comparative review on Apicomplexa nutrient uptake and on transporters which provides it. This can be publish as is. Minor suggestions: ReferencesUse always italic letters for species and genus names also in References.It is unfortunate that all words in titles of the papers start with an uppercase letter as the second word of the species names also start with an uppercase letter (e.g., Toxoplasma Gondii), which looks stupid.

Author Response

We thank the reviewer for taking the time to review our manuscript and for her/his constructive comments.

Point 1: References Use always italic letters for species and genus names also in References. It is unfortunate that all words in titles of the papers start with an uppercase letter as the second word of the species names also start with an uppercase letter (e.g., Toxoplasma Gondii), which looks stupid.

Response 1: We have changed all genus names in italic in the References.

References have been formatted using Zotero setting Microorganisms style.

Reviewer 3 Report

Federica Piro and colleagues present a revision on how Apicomplexa parasites scavenge nutrients from the host. The manuscript is well written and easy to follow. The work represents a valid contribution to the scientific community as Apicomplexa are responsible for several important human and veterinary diseases, and this manuscript summarizes several important points on this pathogens nutrition and may encourage others to pursue further research in the area. The research in parasite nutrition represents a probable target for the development of parasite-selective transport inhibitors, potential contributing to develop new strategies to cure or control parasitic infections.

The manuscript presents two schematic representations of the complex membrane network involved in parasite nutrition helping the reader to follow and visualise the information provided. However, in its present form the captions are not informative to the reader. The figure’s caption should be auto-explanatory for the reader. Please add more information.

Introduction – the introductory section contains very detailed information on several categories of nutrient transporters. Although it is important to contextualize the reader in the different classes of nutrient transporters, I suggest to the authors the creation of separate section dedicated to the description of the several types and classes of nutrient transporters. Even more, I suggest to the authors to add at the end of the introduction, the explanation that due to the lack of studies in other members of the Apicomplexa phylum, the vast majority of information presented in the manuscript was obtained from studies performed in Plasmodium and Toxoplasma, warning the reader to the lack of information on other members of the Apicomplexa phylum.

There is some confusion between the “Introduction” and “Apicomplexa nutrient uptake” section. The Apicomplexa nutrient uptake” section paragraph feels more like the end of the introduction to reader, it contains introductory information, please consider the allocation of this paragraph to the Introduction and the creation of a new section dedicated to the description of the nutrient transporters as described above.

Also, some of the text of the manuscript appear to have some shadow (ex.: abstract, introduction, sections and subsections, abbreviation list). Please remove it.

Table 1 – Please replace it with a higher quality image, as it appears small and out of focus. Please add more information to the table caption, namely discriminate what are PVM, PPM and DV/VAC, for example.

Line 425- There is a “lost” parenthesis. Please remove it.

Line 561- Please remove “conclusion” from the beginning of the phrase as it might be confusing for the reader, as it is not the conclusion section. Replace it for furthermore, overall, for example.

vacuolar compartment  (VAC) is not in the abbreviation list. Please add.

Overall, after the assessment of the above issues, I recommend the publication of the manuscript.

Author Response

We thank the reviewer for taking the time to review our manuscript and for her/his constructive comments.

Point 1: The manuscript presents two schematic representations of the complex membrane network involved in parasite nutrition helping the reader to follow and visualise the information provided. However, in its present form the captions are not informative to the reader. The figure’s caption should be auto-explanatory for the reader. Please add more information.

Response 1: We have added more information on figure’s captions to allow a better understanding of the figures without the support of the text. Moreover, We have enlarged figures to full page to improve their readability.

Point 2: Introduction – the introductory section contains very detailed information on several categories of nutrient transporters. Although it is important to contextualize the reader in the different classes of nutrient transporters, I suggest to the authors the creation of separate section dedicated to the description of the several types and classes of nutrient transporters.  

Response 2: We agree that the description of the different classes of nutrient transporters can make the review too much heavy and in accordance with the suggestion of reviewer 1 we have removed most of these descriptions and generated a new table (Table 1) that summarizes classes and features of transporters.

Point 3: Even more, I suggest to the authors to add at the end of the introduction, the explanation that due to the lack of studies in other members of the Apicomplexa phylum, the vast majority of information presented in the manuscript was obtained from studies performed in Plasmodium and Toxoplasma, warning the reader to the lack of information on other members of the Apicomplexa phylum.

Response 3: We have moved the sentence: “Most of our knowledge on the molecular mechanisms employed by these parasites to scavenge host nutrients is confined to Plasmodium spp. and T. gondii, and thus this review recapitulates and compares these processes in these two important human pathogens.” at the end of the introduction.

Point 4: There is some confusion between the “Introduction” and “Apicomplexa nutrient uptake” section. The Apicomplexa nutrient uptake” section paragraph feels more like the end of the introduction to reader, it contains introductory information, please consider the allocation of this paragraph to the Introduction and the creation of a new section dedicated to the description of the nutrient transporters as described above.

Response 4: We agree with the reviewer that introduction needs to be reorganized.

We have moved the short opening of the section “Apicomplexa nutrient uptake” in the “introduction” section as suggested by the reviewer, that now ends the introduction section. The “Apicomplexa nutrient uptake” section now starts directly with the sub-sections describing solute carriers in Apicomplexa.

Point 5: Also, some of the text of the manuscript appear to have some shadow (ex.: abstract, introduction, sections and subsections, abbreviation list). Please remove it.

Response 5: Shadows have been removed in the revised manuscript.

Point 6: Table 1 – Please replace it with a higher quality image, as it appears small and out of focus. Please add more information to the table caption, namely discriminate what are PVM, PPM and DV/VAC, for example.

Response 6: We have enlarged the Table 2 (previous Table 1) to full page to improve its readability  and increase its quality image.

Furthermore, we have added information on the abbreviations used in the table at the end.

Point 7: Line 425- There is a “lost” parenthesis. Please remove it.

Response 7: We have removed parenthesis in the previous Line 425 (now line 420).

Point 8: Line 561- Please remove “conclusion” from the beginning of the phrase as it might be confusing for the reader, as it is not the conclusion section. Replace it for furthermore, overall, for example.

Response 8: We have replace the word “conclusion” with “overall” in the previous line 561 (now Line 556).

Point 9: vacuolar compartment  (VAC) is not in the abbreviation list. Please add.

Response 9: We have added VAC in the abbreviation list.